# The Finite Element Analysis Research on Microneedle Design Strategy and Transdermal Drug Delivery System

**DOI:** 10.3390/pharmaceutics14081625

**Published:** 2022-08-03

**Authors:** Qinying Yan, Shulin Shen, Yan Wang, Jiaqi Weng, Aiqun Wan, Gensheng Yang, Lili Feng

**Affiliations:** 1College of Pharmaceutical Sciences, Zhejiang University of Technology, Hangzhou 310014, China; yqy@zjut.edu.cn (Q.Y.); 2112007105@zjut.edu.cn (S.S.); 2112007195@zjut.edu.cn (Y.W.); gretta_q@icloud.com (J.W.); 2112107241@zjut.edu.cn (A.W.); 2Department of Ophthalmology, Shanghai Ninth People’s Hospital, Shanghai Jiao Tong University School of Medicine, Shanghai 200011, China

**Keywords:** microneedle, transdermal, personalized drug delivery, finite element analysis, optimization

## Abstract

Microneedles (MNs) as a novel transdermal drug delivery system have shown great potential for therapeutic and disease diagnosis applications by continually providing minimally invasive, portable, cost-effective, high bioavailability, and easy-to-use tools compared to traditional parenteral administrations. However, microneedle transdermal drug delivery is still in its infancy. Many research studies need further in-depth exploration, such as safety, structural characteristics, and drug loading performance evaluation. Finite element analysis (FEA) uses mathematical approximations to simulate real physical systems (geometry and load conditions). It can simplify complex engineering problems to guide the precise preparation and potential industrialization of microneedles, which has attracted extensive attention. This article introduces FEA research for microneedle transdermal drug delivery systems, focusing on microneedle design strategy, skin mechanics models, skin permeability, and the FEA research on drug delivery by MNs.

## 1. Introduction

Microneedles (MNs), as a non-invasive alternative to oral and I.V. (intravenous injections) administration, can be considered an invaluable route to bypass the stratum corneum (SC) barrier and avoid the gastrointestinal and hepatic first-pass metabolism [1]. Thus, MNs can combine with all carriers, including nanoparticles, microspheres, and liposomes, regardless of the molecular weight of the drug, for applications such as beauty, vaccines, genes or proteins, and hydrophilic drugs [2,3,4,5,6,7,8,9,10]. In addition, the MNs have been deemed ideal for biosensing and have been explored as routine point-of-care health monitoring devices for transdermal detection of cancer biomarkers or physiologically relevant analytes [11,12]. The MNs can be designed and fabricated into various shapes from different biocompatible matrix materials. Given the unique structural and mechanical properties of biocompatible matrix materials, the design and manufacturing methods of MNs have always been widely discussed [13]. The current preparation methods [14,15,16,17] of polymer microneedle arrays include micro-molding, drawing techniques, 3D printing, etc. Due to the complex production process, the consumption of raw materials and expensive equipment hinders the development from laboratory research to industrial production [18]. The critical mission for MNs is to ensure human safety and improve efficacy during transdermal drug delivery. In addition, many significant challenges still need to be solved, including the problem of repeated penetration when acting on human skin, the improvement of drug loading methods, and the problem of controlling the orientation of drug deposition to the layered target area of skin tissue [19]. The clinical challenges of microneedle technology are as follows: (1) The pre-screening and prescription optimization of microneedle materials is time-consuming, and the problem of low-cost industrial manufacturing urgently needs to be solved; (2) the actual drug dose varies depending on the penetrating depth of the skin. Hence, designing individualized microneedles that can accurately control the administered dose remains a huge challenge; (3) the limitations of the microneedle’s small drug delivery capacity prevent the effective dose for clinical administration for many drugs; and (4) at present, there is still a lack of a unified standard system for the quality evaluation of microneedle products.

The past decade has seen the rapid development of finite element analysis (FEA) in many research fields, especially for medical purposes [20], which has been widely used in many areas of biomechanics [21,22,23,24,25], such as cardiovascular, orthopedics, eye, and brain, etc., as shown in Figure 1. The FEA uses digital and mathematical modeling to make an approximate solution for each element based on the known number of nodes, the coordinate system of each node, and the material characteristics, and finally derive the total solution for this quantitative domain analysis to quantify the actual complex problems [26]. As shown in Figure 2, specific steps are as follows: (1) Establish an approximate research object model; (2) divide the research object into a limited number of units, assign material properties, loads, and impose boundary conditions; (3) using standard methods to propose an approximate solution for each unit, the researcher can quickly analyze the behavior of the basic unit and propose a method for solving the basic unit; (4) combine all the units into a system similar to the original system according to the standard method, and assemble the basic units into an approximate system, approximately representing the research object in terms of geometry and performance characteristics; and (5) numerically solve this approximate system [21]. It can then be simulated in a non-linear finite element solver to estimate the internal stress, strain, and deformation under load.

In recent years, the FEA has attracted considerable attention in the field of MNs. It is expected to be a prospective application in MN development, eliminating the need for many time-consuming and expensive experimental trials [27] and making personalized MNs according to the different skin parameters of various patients [25]. The FEA has attracted considerable attention in human injury biomechanics. This article mainly introduces studies on the FEA as an auxiliary method for MNs. Firstly, the MNs design strategy of FEA simulation is introduced. Secondly, literature on skin mechanics behavior and FEA skin models are covered. Thirdly, the drug transport processes and in vivo processes resulting from MN insertion are discussed. Finally, the challenges of the present and visions for the future of FEA are also summarized.

## 2. MNs Design Strategy by FEA

### 2.1. Based on the Matrix Materials of MNs

In general, MNs can be made of a wide variety of materials, such as silicon, metal, titanium, glass, and polymers [31]. Each material has its advantages and disadvantages. The properties of materials directly affect the preparation of MNs, skin penetration, and drug release [32,33,34]. At the same time, the compatibility of drugs and materials, preparation process, application and the type of administration will also affect the choice of MNs materials [35]. Compared to silicon-like brittle materials, polymer materials have higher biocompatibility for their ability to avoid brittle breakage when penetrating the skin or other tissue, which was considered the most promising MN manufacturing material. For polymer MNs, the degradable polymer materials can be chosen according to the drug degradation rate to control the release characteristics of the drug [36]. However, there are also some problems concerning polymer MNs. For example, some polymers are soft and do not have enough mechanical strength, which could cause a catastrophic buckling failure of the MNs during the penetration.

To characterize the mechanical properties of materials, many studies used devices such as texture analyzers [37], micromechanical testing machines [38], displacement force testing machines [39], and nanoindenters [40] to investigate the mechanical strength of MNs as well as verification of whether the MNs can successfully puncture the skin and achieve effective transdermal delivery of micro-targeted drugs. The micromanipulation was proposed by Du et al. to directly and accurately measure the breaking behavior of a single microneedle and provide information about the uniformity of the microneedle strength of the entire patch, which provided a method to characterize the mechanical strength of MNs [41]. Wang et al. [42] used a micro-mechanical test machine to evaluate the mechanical strength and compare the influence of relative humidity and other factors on the mechanical properties of MNs. Zhang et al. [43] used nanoindentation technology to obtain the displacement load curve, its elastic modulus, and a hardness histogram to evaluate the mechanical strength of the material. However, it cannot realistically assess the mechanical strength of the MNs, particularly to reflect the MNs’ penetrating the human skin.

In recent years, finite element analysis (FEA) has been used to test and verify the mechanic properties of polymer materials of MNs and complement the experimental findings, which attracted extensive attention. Eriketi Z. Loizidou et al. [44] performed experimentally a FEA to study the mechanical properties of sugar MNs. The buckling force and von Mises stresses were the index for predicting MN failure. A certain correlation between the Young’s modulus of the material and the predicted microneedle critical flexion load and the depth of skin penetration was confirmed. Urvi Kanakaraj et al. [45] performed the structural analysis of 10 materials based on the buckling and bending forces using COMSOL Multiphysics. The current parameters of microneedle material are mainly Young’s modulus and Poisson ratio, and the material is set as a linear elastic material. However, the property of the material is the result of a combination of multiple parameters, and the simple Young’s modulus and Poisson ratio cannot well reflect the mechanical strength of the MNs. To facilitate model convergence, some scholars see MNs as simple analytical rigid bodies, which often ignore the deformation of the material [46]. Table 1 lists the mechanical parameters and features of common microneedle matrix materials, which provide a basis for subsequent FEA simulations. The mechanical strength, elastic modulus, and fracture toughness reflect the insertion ability of the polymer-based MNs [47]. Evaluating the mechanical properties of polymer MNs directly and accurately, especially in the case of industrialized mass production, is necessary to ensure their successful application.

### 2.2. Based on the Morphology of MNs

According to the processing techniques, there are two types: in-plane and out-of-plane MNs [51]. MNs can be fabricated in different forms: solid, coated, soluble, and hollow. In recent years, people have been committed to developing different forms of MNs and a new generation of smart MNs to meet different application scenarios (such as wearable devices, drug response, drug delivery, testing, etc.). In the process of penetrating the MNs into the skin, the construction of the MN structure will withstand external forces such as lateral pressure, axial pressure, shear force, and frictional force. On the micron scale of MNs, a small amount of force may cause them to break, deform, or exhibit other failure modes. Moreover, there are nerve endings in the tissue layer below the dermis, and the human body will feel pain once touched. At present, the transdermal drug delivery technology based on MNs is mainly focused on how to manufacture the effective structure of the MNs itself, to better penetrate the skin without breaking, and achieve better drug delivery. Reasonable modeling of MN structures has become a research hotspot in transdermal drug delivery systems. Many scholars have attempted to develop FEA to predict the effect of different MN parameters on the mechanical properties of MNs [52], such as geometrical size [53], type (cone, tapered-cone, beveled-cone, pyramid) [54] (Figure 3), shape parameters (tip area, wall angle, wall thickness) [46], density of the MNs [55,56], and the geometry of the MN base [57]. Loizidou et al. [57] investigated how the geometric composition of MNs affects their mechanical strength and penetration characteristics by simulating MNs with triangular, square, and hexagonal base geometries. The average von Mises stress and critical buckling load were determined to evaluate the mechanical properties of MNs, as shown in Figure 4.

Drug molecules often enter the blood circulation through passive diffusion or combine with ion penetration to accelerate their diffusion efficiency. Transportation efficiency is mainly related to the size and depth of the channel [58]. The design of MNs of various sizes can produce different changes in the mechanisms and effects of drug delivery [59]. Among all types of MNs, hollow MNs can transport or extract biofluids in a controllable manner. However, because of its structure and fragility, the manufacturing may become more complicated than that of solid MNs, and there is a very high possibility of blockage problems. Ahmad, N.N. et al. [60] proposed a hollow side-open and outer-grooved design of MNs, of which the effects on the skin puncture properties were investigated, as shown in Figure 5. Mechanical structural analysis and hydrodynamic analysis were used to verify the designed MNs structure. The presence of the groove can reduce the contact interactions and thus reduce the insertion force. The insulin can be delivered in an ultrafast manner with the assistance of a capillary pressure induced by the outer-groove structure of the MNs. Jennifer García et al. [61] presented a static analysis with the computational fluid dynamic (CFD) tool to evaluate the von Mises stress of the fluid passing through the MNs and the total deformation during the simulation process. Wei Yafei [62] designed MNs with ultra-sharp star and side openings to analyze the structural strength, stiffness, and buckling stability of the needle tips using FEA tools. At the initial penetration stage, the pressure is concentrated on the tip area of the MNs. After piercing the skin, the force of the MNs is mainly focused on the joints of the MNs. This particular structure not only reduces the required penetration force but also effectively improves the transmission efficiency of the liquid medicine in the transdermal delivery process. Similar research was also conducted by Xenikakis, I. [63] and Jiaming Chen [64].

Many scholars have tried to improve the mechanical characteristics and drug-release performance of MNs by designing bio-inspired MNs. Inspired by the insertion mode of the mosquito, Aoyagi et al. [65] designed the combined MNs composed of a central straight needle and two outer jagged ones to investigate the effect of the insertion approach and the effectiveness of the cooperative motion of these needles. The FEA results revealed that the stress distribution could be confined to the space between two maxillae. The degree of stress concentration gradually increased, which considerably reduced the insertion force and penetration difficulty. Zhipeng Chen et al. [66] created a honeybee-inspired microneedle to explore whether bionic MNs with different structures and barbless would affect the adhesion and stress when penetrating the skin. The FEA results confirmed that the bionic design exhibited a smaller penetration force and a greater adhesion force.

## 3. Characteristics of Skin Mechanics and FEA Models

The skin is mainly composed of the epidermis, dermis, and subcutaneous tissue. Since the structure and tissue composition of each layer are different [67], as shown in Figure 6, the corresponding mechanical properties are significantly different. The epidermal layer is located in the outermost layer of the skin, with a thickness of 20~150 μm [68]. It forms a protective barrier on the body surface, responsible for maintaining water in the body and preventing the invasion of pathogens [69]. The dermis is below the epidermis, with a thickness of 150 μm~4 mm, composed of connective tissue and rich in collagen fibers, elastic fibers, and reticular fibers [70]. They are intertwined into a net to make the skin more elastic and tougher, as the main factor to guarantee the mechanical properties of the skin. The subcutaneous tissue has the functions of buffering mechanical pressure, storing energy, and keeping warm, with poor resist deformation ability, which can be regarded as a body filling between the skin and bone [71]. Experimental research methods have been extensively performed in previous studies to evaluate the mechanical properties of skin. Skin mechanical behavior has also been previously estimated in a multitude of conditions, by cyclic loading-unloading tests [72], rupture tests [73], and in vivo tests [74]. After a mass of skin mechanics tests, various mechanical properties of skin are derived [75], such as anisotropy, Young’s modulus, stiffness, compressibility, strength, toughness, initial stress, and skin friction coefficient.

The stiffness and strength of skin tissue are affected by many factors such as the external environment (humidity, temperature, etc. [77]), different parts of the human body, gender, age, and race. The biological nature of the species, location, orientation, and sex may all introduce variations to skin mechanical properties [78]. When subjected to mechanical stimuli such as friction and pressure for a long time, the skin stratum corneum will thicken, which will increase the local anti-pressure and friction resistance, such as toes, knees, and palms [79]. As age increases, the composition of elastic fibers in the skin decreases, accompanied by a function decline. In addition, the condition of skin health also affects the corresponding skin mechanical properties [80]. In conclusion, the biomechanical properties of skin are a complex result of the interaction of different layers (epidermis, dermis, and subcutaneous tissue) [81], which cannot be elucidated entirely and accurately so far. However, approximate profiles of certain properties have been obtained by studying elasticity, viscosity, and plasticity [82].

Understanding the maximum stress skin tissue can withstand is an essential precondition for MN design. In recent years, plenty of classic models have been used to describe the mechanical properties of the skin [83,84,85], from an early empirical model to a more scientific structural model. Compared to traditional in vivo and in vitro animal models, the constitutive model based on FEA can fit the experimental results well by correlating constitutive parameters with meaningful physiological values [86]. For instance, Chen Sheng et al. [78] developed an automatic image analysis program and measured the distribution of relative collagen fiber bundle orientation through histological images. A microstructurally based constitutive model was proposed to characterize the non-linear anisotropic mechanical behavior of the skin by integrating mechanical parameters, constitutive model, and collagen microstructure data. The current ordinary finite element multi-layer skin model and corresponding parameters were summarized and compared, as shown in Table 2.

## 4. Skin Permeability and the Drug Transport Processes

Permeability is another important property of the skin. The penetration and absorption of the skin are mainly conducted through two pathways: the skin appendages and the stratum corneum. Skin appendages combined with hair follicles, sebaceous glands, and sweat glands account for only 0.1% of the skin area. Ions or macromolecules can be absorbed through this pathway [93]. However, it is generally assumed that the major barrier to transdermal penetration is the stratum corneum [94]. Due to the complex skin structure and limited experimental facilities, the drug permeability of different skin layers cannot be obtained experimentally. The predictive mathematical model for skin penetration was developed from the steady-state models (quantitative structure-permeation relationship models, structure-based models, and porous pathway models) to the transient models with time dependence (including basic models, compartment models, complex models, and slow binding/partitioning kinetics in the SC), in which the compartment models, also known as PK models, could be applied to trace the drug fate after penetration into the skin [95].

Mathematical modeling of epidermal and dermal transport is essential for the optimization and development of products for transdermal delivery, especially for the optimization of microneedle transdermal drug delivery systems. Fortunately, FEA meets this requirement. For example, FEA can be used to decompose the complex physical model into disordered structures and heterogeneous media, and then vary the density of the mesh to simulate the diffusion properties of the skin [96]. Transcellular and lateral lipid diffusion pathways were modeled within a brick-and-mortar geometry representing fully hydrated human SC by Barbero, which could be used to gain insight into the stratum corneum (SC) permeation pathway for hydrophilic compounds. The review discussed and summarized published models concerning skin permeation using FEM [97]. According to the application (scale and variables), the models are grouped as macroscopic (multilayered slabs), microscopic (unit cell), and macroscopic with microscopic details [96]. Calcutt et al. [98] recently published a review summarizing the models that explain the drug transport within the viable skin and stratum corneum, microneedle dynamics, and estimation of the diffusion coefficient. Yongwei Gu et al. [99] set an FEA model of the multi-layer skin to simulate the dynamic permeation process of paeonol nanoemulsion (PAE-NEs) through the skin and conducted in vivo experiments, as shown in Figure 7. The skin pharmacokinetic characteristics of PAE-NEs obtained by simulation are consistent with the experiments in vivo, which intuitively demonstrated the drug transmission process over time and skin depth. Rim et al. [100] simulated two-dimensional (axisymmetric) drug diffusion from a finite element drug reservoir into the skin by FEA. They established a finite element formula for transient multi-component non-linear diffusion in layered media and modeled the skin as a uniform continuous domain. The formula considered the possibility of compound distribution and penetration enhancement between different domains, which presented a universal characteristic that was uniquely suited for simulating linear and non-linear, single-component, and multi-component diffusion problems. Khanday et al. [101] established a variational FEA with a linear shape function and used the Lagrange interpolation method to calculate the drug concentrations at different dermal nodes. Römgens et al. [102] used a combination of fluorescent recovery after photobleaching experiments and FEA to study the diffusion coefficients of two fluorescent glucan molecules with different molecular weights in the epidermis, papillary dermis, and reticular dermis. In addition, the algorithm can deal with complex geometric conditions that are difficult to deal with by classical analytical solutions.

It can be concluded that the method of FEA based on a multi-layer geometry model is a promising strategy to predict the TDDS skin pharmacokinetics as well as to quantitatively describe the absorption and distribution behavior of the drug after transdermal administration. Nevertheless, many parameters required for advanced modeling are not readily available, which remains a considerable challenge. Improving the modeling parameters through experiments and establishing a more efficient model are all of great value for the in-depth study.

## 5. The FEA Research on Drug Delivery by MNs

After MNs penetrate the skin, their quality, safety, permeability, and biocompatibility are essential issues that must be clarified before the clinical application of MNs [13]. Plenty of new approaches by imaging technology for monitoring the effect of the release of the microneedle administration are being explored steadily, such as the confocal microscopy method [103], two-photon microscopy method [104] , and optical coherence tomography (OCT) method [105]. OCT is a non-invasive, high-quality imaging method that can observe the contour and failure state of the microneedle in the penetrating process and the dissolution process over time [106]. It is worth noting that developing a high-resolution, sizeable focal depth OCT system is currently a major challenge. The purpose of making MNs is to better deliver the drugs encapsulated in the MNs and deliver them to the target site. To evaluate whether the drugs in the MNs are released, the rate of release, and the amount of release, scientists utilize in vivo or in vitro methods to determine it. The in vivo method primarily applied the prepared MNs to live animal skin and analyzed the drug content of MNs by collecting blood or skin. The in vitro methods include the Franz diffusion cell and microdialysis, etc. At present, microneedle transdermal drug delivery technology is still in its infancy, with a lack of a unified standard system for evaluating the quality of microneedle products.

Compared with traditional biomechanical methods, for instance, animal experiments, physical experiments, and in vitro experiments, the mathematical models and FEA methods constitute a powerful predictive tool to better understand the matrix degradation and drug release of the MNs. For solid MNs, the drug can be loaded into the microneedle matrix. In non-degradable ones, drugs with low MW will diffuse to the outer medium, while in degradable matrices, drugs with higher MW can be released as the polymeric matrix degrades [107]. Benslimane et al. [108] carried out a mathematical model to investigate drug diffusion from transdermal drug delivery using MNs. Fick’s second law differential equation is solved numerically using the well-known finite difference method, and a closed form of an exact solution is obtained to lead to a concentration field. The study explored the effect of the diffusion coefficient, initial concentration, and the length of MNs on diffusion and concentration over time and space of transdermal drug sustained delivery. Barrak et al. [109] carried out a numerical model to study the pharmacokinetics of drugs administrated by MNs. This algorithm provided a valuable tool for the drug delivery performance evaluation of MNs. Lyashko et al. [110] developed the finite-difference methods and a two-step symmetrizable algorithm to predict the best drug concentration (DC) distribution and get the best model for dissolving microneedles. Machekposhti et al. [111] simulated the diffusion of tranexamic acid (TXA) delivered by MNs in the skin layer using the FEA software COMSOL 5.0. According to Fick’s diffusion law, the DC of TXA in plasma (interstitial fluid) was determined, and the effective DC in the skin was then calculated from the known skin porosity. The simulation of drug diffusion in skin layers at 0 h, 2 h, 12 h, 24 h, 2 days, and 4 days after polymer microneedle insertion was shown in Figure 8. Zoudani et al. [112] built a model of drug release from single dissolving MNs in a controlled volume to estimate the drug concentration in the skin layer and the dissolution process. The effects of the initial drug load, pitch size on the dissolution rate, and drug concentration in the tissue were discussed. Later, a novel microneedle shape with a hemispherical convexity array was designed for further simulation. Castilla-Casadiego [113] used COMSOL Multiphysics 5.4 software under the physics of transport of diluted species and time-dependent study in a space dimension of 3D to investigate transdermal drug delivery. Laminar flow was placed on the fat layer to simulate capillary blood flow. The results demonstrated that the transdermal drug delivery efficiency increased with the number of microneedles on the surface patch and the percentage of penetration depth. To sum up, FEA provides an economical and effective method for studying the drug delivery process and explaining the transdermal mechanism of MNs.

## 6. Conclusions and Outlook

Despite the extensive research and advances in the field of transdermal drug delivery, there are still many technical issues related to microneedle drug delivery systems, including their mechanical strength, biocompatibility, dose limits, dose accuracy, and application methods [114]. Pen-type and syringe-type microneedle systems have been commercially approved, but their development seems to have stalled. A few MNs products are currently on the market, and many clinical trial results have not yet been published, which is worth pondering. There is an urgent need to establish improved indicators for the evaluation of MNs to drive commercialization. In addition, the pre-screening and prescription of MNs were time-consuming, and the biomechanical properties of the skin are the consequence of a complex interaction of various species, sexes, sites (epidermis, dermis, and subcutaneous tissues), different environments, and disease states. In vitro and in vivo animal studies are not the best options for evaluating transdermal drug delivery, which also limits the movement of microneedle technology from trials to clinics. Once the relevant problems are solved, it is expected that the preparation technology bottleneck will be broken, and the productive application will be extended drastically for microneedles.

Along with biological simulations, the FEA can estimate biomechanical responses and the dynamic process of microneedle insertion. The FEA of simulated insertion and fracture forces can be used to understand the biomechanics of microneedles. The FEA plays an important role in MNs’ design by considering these categories. Compared with studies on real-life animal models, FEA as a computer-based design method can assist microneedle preparation techniques because they can be modified and altered to meet specific requirements and provided on an individual basis without ethical considerations, which can save a great deal of time [115], from the perspective of microneedle mechanical properties. A specific correlation between experimental and simulation was confirmed [116]. For the mechanical properties, the buckling force, von Mises stresses, and penetration force have been proved to predict the quality of microneedles. And the geometry plays a more critical role in the mechanical properties of microneedles as well as drug-release performances. The FEA showed promise in predicting the mechanical properties of microneedles. From the perspective of microneedle drug release performance. The FEA methods constitute a powerful predictive tool to understand matrix degradation and drug release from MNs by simulating the drug delivery process in the skin, which can serve as a guide for the preparation of individual microneedles for precise and accurate drug delivery. From the perspective of in vitro and in vivo animal studies’ limitations, the FEA can assist in vitro experiments and predict in vivo behavior to better solve the current problems of microneedle in clinical practice.

Nevertheless, the model is not entirely realistic and may contain inaccuracies in segmentation, meshing, or biomechanical material variables [117]. Although a large number of skin models have been developed for the design of microneedle drug delivery systems, the available skin models are often simplified, such as by simplifying skin damage. For understanding the biological behavior of MNs and forces of action in transdermal drug delivery, it is often challenging to establish a reasonable skin model that can reflect its biology to the greatest extent possible. Although complex models can increase the realism of the results, their slow and time-consuming operation is a major limitation. In addition, model validity tests are probably verified in vitro rather than in vivo due to limited experimental conditions, which needs further improvement. Moreover, the model of the FEA simulation is based on the raw data, from which the accuracy of the data acquisition, the structure of the modeling, and the material authenticity are all the key factors affecting the effectiveness. On the other hand, the material properties and parameters of the FEA have mostly referenced documents, which are greatly different from their experimental conditions. It is worth noting that FEA is still in its infancy with limitations, which means it cannot completely replicate the actual clinical state to the extent or truly reflect the real penetration situations and still lacks the verification standard.

In this paper, the applications of FEA in MN design strategy, skin mechanics characteristics, skin permeability, and drug delivery by MNs were reviewed, indicating that FEA could be useful in the study of microneedle transdermal drug delivery systems, as well as the design and mechanical properties testing of MNs. Before the preparation of MNs, the stress, penetration, and buckling of different types of MNs can be designed and predicted, which can avoid ineffective design. For the prepared MNs, the experimental results can be supplemented and verified using FEA simulation to better understand the properties of MNs. After administration of MNs, the drug dissolution, release characteristics, and skin pharmacokinetics can also be predicted. However, the following two main problems should be focused on and solved in the future: (1) How to obtain more realistic skin and MN parameters using the existing experimental method and (2) how to establish a comprehensive FEA method platform for microneedle delivery systems with the concept of “design MNs by computer simulation, calibrate the simulation parameters by experimental data, and support the experimental results by quantitative analysis.”

We believe that the above problems will be solved through the deepening of research on transdermal drug delivery systems and the further improvement of microneedle preparation technology. In addition, an optimized simulation based on the experimental results can explain the other experimental data and ultimately guide the experiment design. It can be foreseen that FEA simulation will play an increasingly important role in the development of MNs TDDS, as depicted in Figure 9, and will have a bright future in the design and research of personalized and intelligent drug delivery systems.

## Figures and Tables

**Figure 1 pharmaceutics-14-01625-f001:**
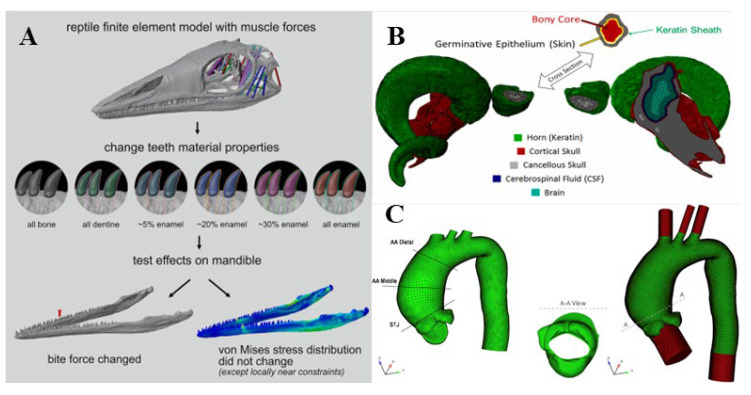
Application of FEA methods in the field of biomechanics. (**A**) tooth enamel models [28], (**B**) brain model during impact (Reprinted with permission from Ref. [29]. Copyright 2021, Elsevier), (**C**) ascending thoracic aortic aneurysms models (Reprinted with permission from Ref. [30]. Copyright 2018, Elsevier).

**Figure 2 pharmaceutics-14-01625-f002:**
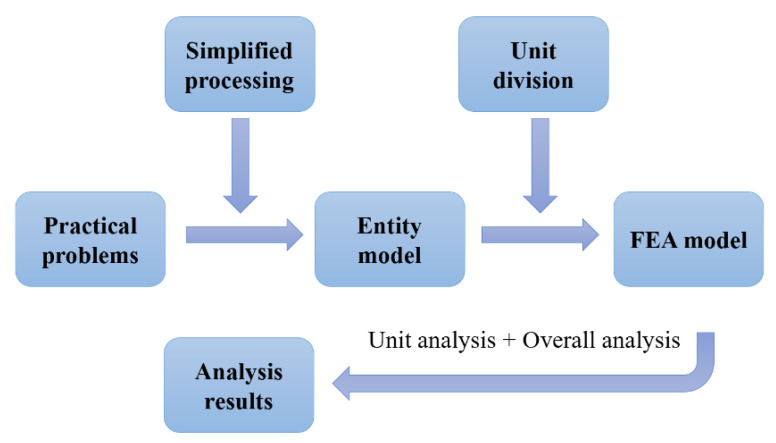
The finite element analysis process.

**Figure 3 pharmaceutics-14-01625-f003:**
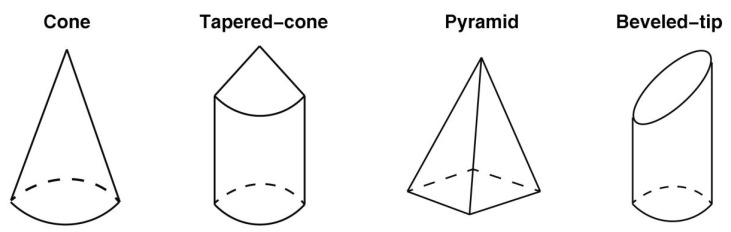
The different types and sizes of microneedle models [54].

**Figure 4 pharmaceutics-14-01625-f004:**
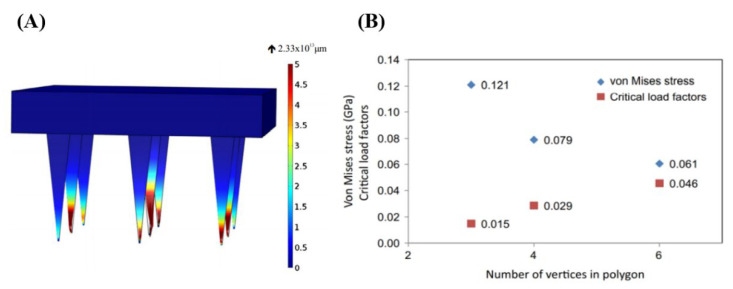
(**A**) Buckling modes of a triangular-shaped base, (**B**) von Mises stress and critical load factors of MNs with triangle, square, and hexagon base geometries (Reprinted with permission from Ref. [57]. Copyright 2016, Elsevier).

**Figure 5 pharmaceutics-14-01625-f005:**
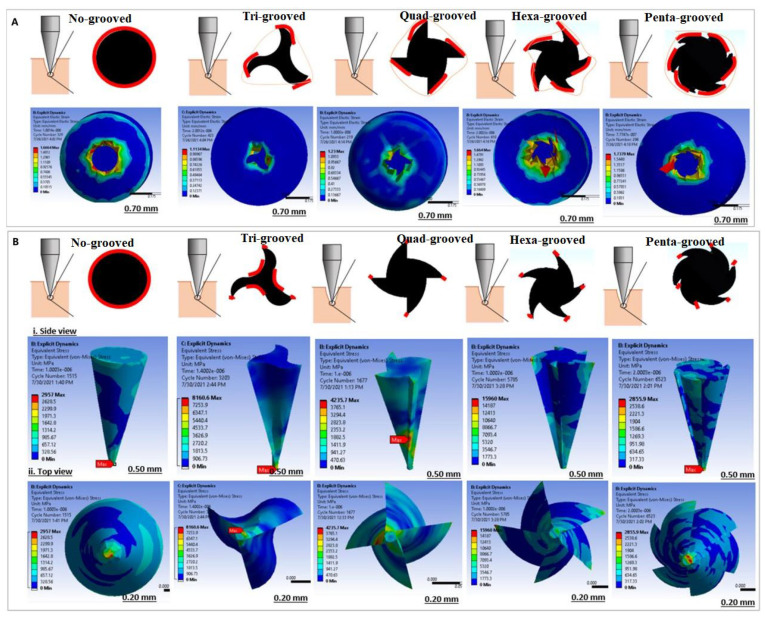
Strain (**A**) and Stress; (**B**) distribution of variable outer-grooved design (Reprinted with permission from Ref. [60]. Copyright 2021, Elsevier).

**Figure 6 pharmaceutics-14-01625-f006:**
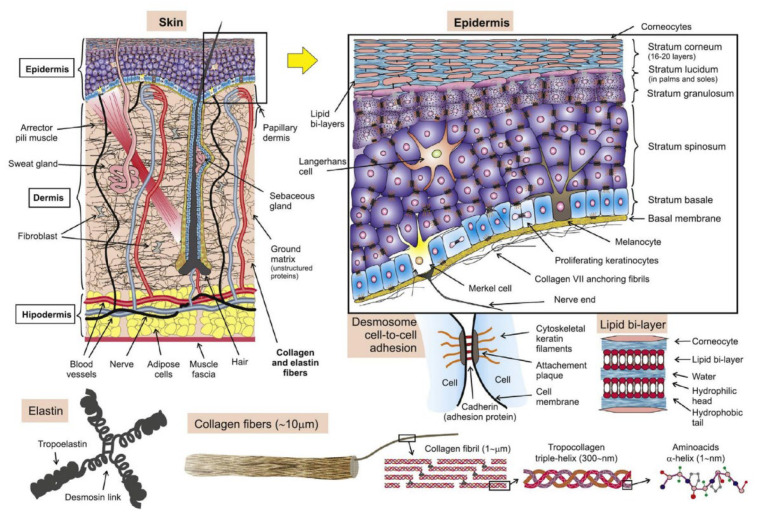
The structures of the skin (Reprinted with permission from Ref. [76]. Copyright 2017, Elsevier).

**Figure 7 pharmaceutics-14-01625-f007:**
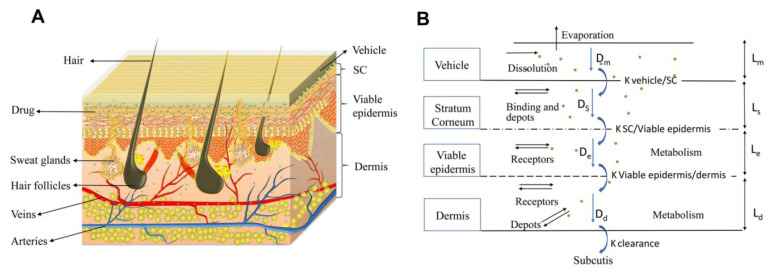
The multi-layer geometry model of the drug transport system: vehicle, SC, viable epidermis, and dermis. (**A**) The 3D schematic diagram of vehicle and skin texture. (**B**) The 2D schematic diagram of drug permeation process from vehicle to subcutis and the highlighted pathways remarked with diffusion coefficients (D_m_, D_s_, D_e_, and D_d_) [99].

**Figure 8 pharmaceutics-14-01625-f008:**
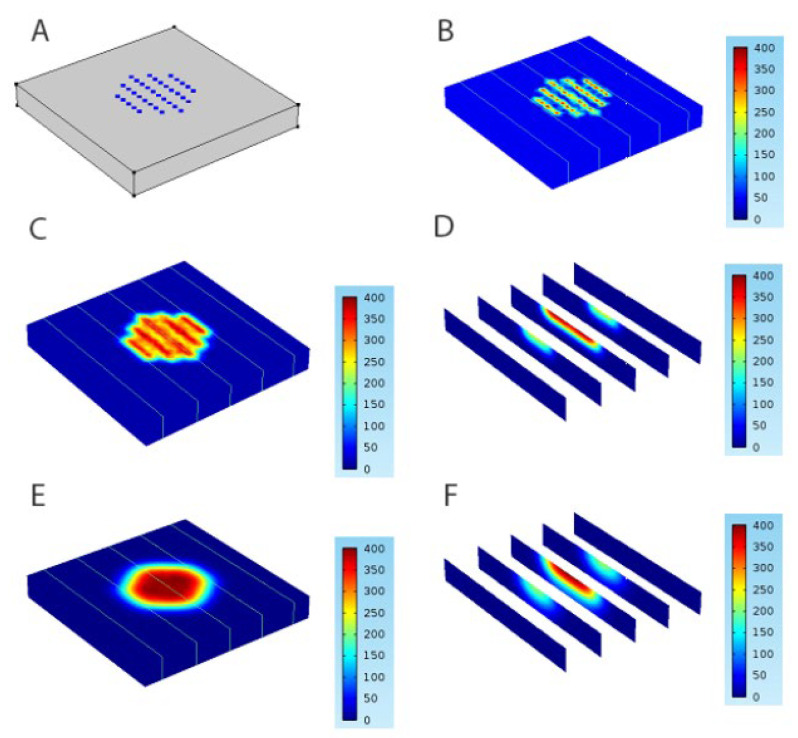
The FEA of drug diffusion in skin layers in (**A**) 0 h, (**B**) 2 h, (**C**) 12 h, (**D**) 24 h, (**E**) 2 days, and (**F**) 4 days after polymer microneedle insertion (Reprinted with permission from Ref. [111]. Copyright 2017, Elsevier).

**Figure 9 pharmaceutics-14-01625-f009:**
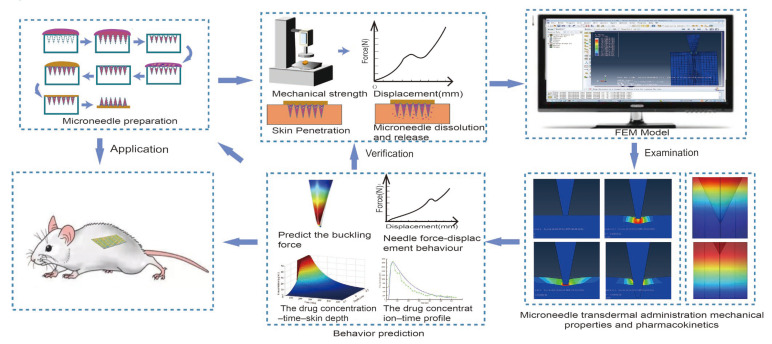
Conceptual diagram of FEA simulation for MNs TDDS.

**Table 1 pharmaceutics-14-01625-t001:** The mechanical parameters and features of common microneedle matrix materials [41,44,48,49,50].

Microneedle Material	Density ρ [kg/m^3^]	Young’s Modulus E [GPa]	Poisson’s Ratio ν	Yield Strength [GPa]	Characteristic
Silicon	2329	170	0.28	7	Brittle materials with good stiffness, hardness, and biocompatibility
Polysilicon	2320	169	0.22	7	High strength, acid and alkali resistance, high-temperature resistance
Silicon Carbide	3216	748	0.45	21	Anti-oxidation, low thermal expansion, erosion resistance, corrosion resistance, low density, high strength, high modulus, wear resistance
Borosilicate glass	2230	66.3	0.22	3.6	Good mechanical properties
Titanium	4506	115.7	0.321	0.1625	Low cost, excellent mechanical properties
Steel	7850	200	0.33	0.250	Has excellent comprehensive mechanical properties, easily broken and left in the body
Silk	1340	8.55	0.4	0.500	Has excellent toughness and ductility
Maltose	1812	7.42	0.3	7.44	Very common excipient in FDA-approved parenteral formulations, the most commonly used sugar for preparation of MNs, easily absorbs moisture
Polycarbonate (PC)	1210	2.4	0.37	0.070	Good biodegradability and biocompatibility
Polyurethane (PU)	1120	0.055	0.39	0.000196	High abrasion resistance, low-temperature capability, ambient curing, and comparatively low cost
Polyvinyl pyrrolidone 58 (PVP 58)	1062	2.4	/	/	Too brittle
Polylactic acid (PLA)	1251.5	1.280	0.36	0.05345	Higher modulus of elasticity
Poly-L-Glutamic Acid (PGA)	1530	9.9 ± 0.3	0.3	0.09	Has a higher modulus of elasticity
Poly Lactic-co-Glycolic Acid (PLGA)	1000	3	/	0.05	Combined with other quick-release materials in different ways to achieve various purposes

**Table 2 pharmaceutics-14-01625-t002:** Different skin constitutive models (Reprinted with permission from Ref. [87]. Copyright 2010, Elsevier. Reprinted with permission from Ref. [88]. Copyright 2017, Elsevier).

Model Diagram	Constitutive Model	Material Parameters	Positives and Negatives	Ref.
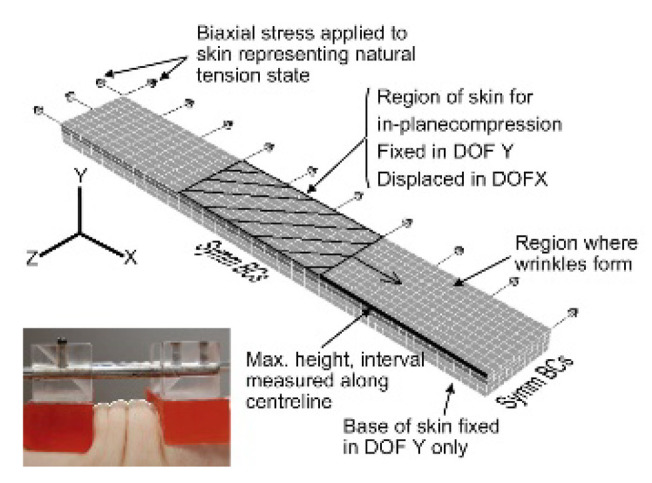	Stratum corneum: Isotropic Neo-Hookean $W=C_{10}$$(I_{1}-3)+\;\frac{1}{D_{1}}$(J − 1) Dermis: W = $W_{X0}+\frac{\eta X\kappa \theta }{4}$$[N\;\times \;\sum_{i}^{4}(\frac{\rho_{X}^{\left(i\right)}}{N_{X}}\beta_{\rho_{X}}^{\left(i\right)}$$+\ln\frac{\beta_{PX}^{\left(i\right)}}{sinh\beta_{PX}^{\left(i\right)}}$$)-\frac{\beta_{PX}}{P_{X}}\ln(\lambda_{a_{X}}^{a_{X}^{2}}\lambda_{b_{X}}^{b_{X}^{2}}\lambda_{c_{X}}^{c_{X}^{2}}$)] + B{cosh(J_X_ − 1) − 1} Hypodermis: Isotropic Hyperelastic Yeoh W = $C_{10}(I_{1}-3)+C_{20}(I_{1}-3{)}^{2}+C_{30}(I_{1}-3{)}^{3}+\frac{1}{D_{1}}$$(J-1{)}^{2}+\frac{1}{D_{2}}$$(J-1{)}^{4}+\frac{1}{D_{3}}$(J − 1)^6^	Stratum corneum: Relative humidities (RH): 30%, 75%, 85%, 92%, 96%, 100% Young’s modulus, E(MPa): 960, 240, /, /, /, 5–6 C_10_(MPa): 160, 40, 24, 12, 4, 1 D_1_(MPa): 0.00025, 0.00101, 0.00169, 0.00338, 0.01013, 0.0405 Hypodermis: C_10_(KPa): 1.649, C_20_(KPa): −1.136, C_30_(KPa): −1.792	It can replicate the changes in skin layers with different properties and different inherent tension in the process of wrinkle formation. However, since the surface of real skin is not perfectly smooth and the surface where wrinkles form is not flat, the model is just a simplification of real skin.	[87]
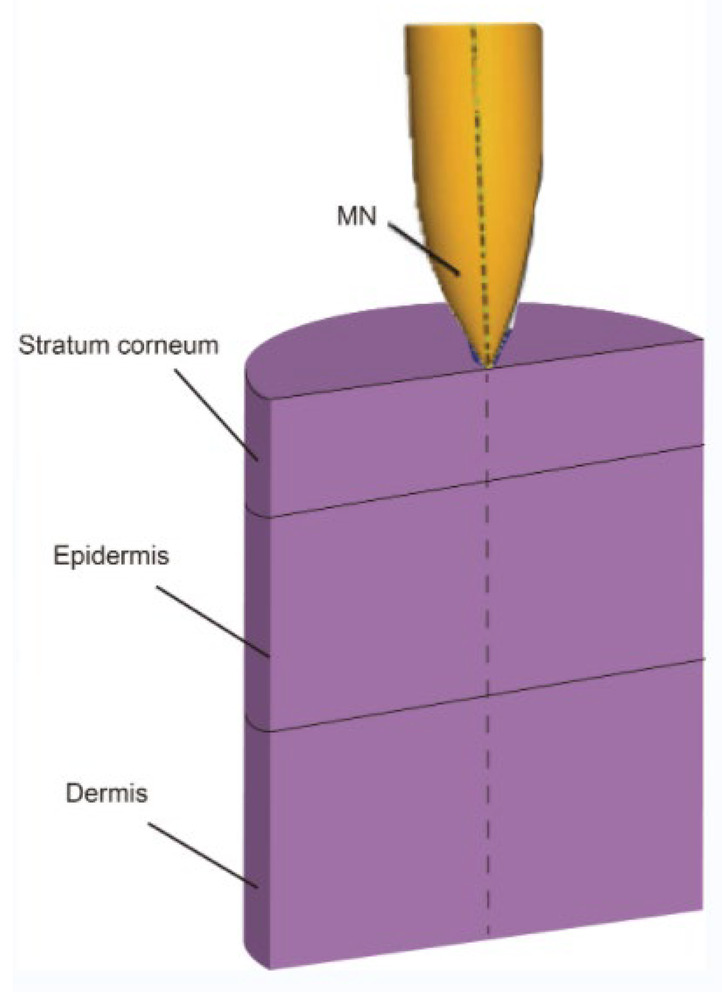	Stratum corneum: Isotropic Neo-Hooken $U=C_{10}$(I_1_ − 3) Dermis: Isotropic Neo-Hooken $U=C_{10}$(I_1_ − 3) Hypodermis: Elastic	Stratum corneum: $C_{10}(\mathrm{MPa}):\;10;\;\sigma \left(\mathrm{MPa}\right)$: 37 Dermis: $C_{10}(\mathrm{MPa}):\;0.2;\;\sigma \left(\mathrm{MPa}\right)$: 7 Hypodermis: Young’modulus, E(Pa): 3.4 × 10^4^, υ: 0.48	It can successfully predict the deformation and damage of multi-layer skin and the penetration force of micro acupuncture into the skin. However, the skin failure model requires programming in a subroutine.	[89]
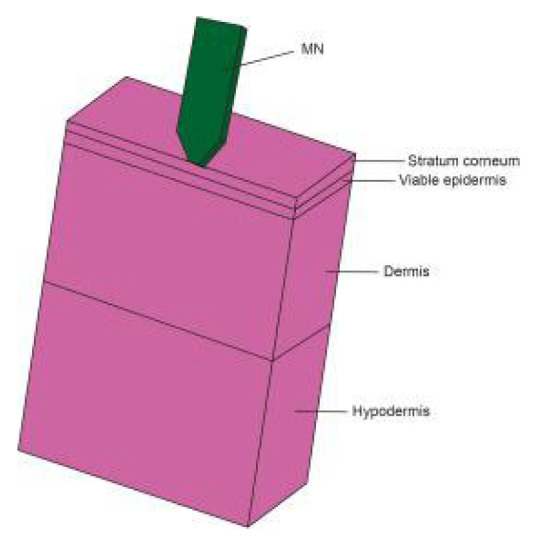	Stratum corneum: Hyperelastic Ogden model (α = 8.68) $\Phi =\left(\frac{2\mu }{\alpha }\right)(\lambda_{1}^{\alpha }$$+\lambda_{2}^{\alpha }$$+\lambda_{3}^{\alpha }$$)+d\;\times \;f(\lambda_{1}$$,\;\lambda_{2}$$,\;\lambda_{3}$ Viable epidermis: Hyperelastic Ogden model (α = 20.68) $\Phi =\left(2\mu /\alpha \right)(\lambda_{1}^{\alpha }$$+\lambda_{2}^{\alpha }$$+\lambda_{3}^{\alpha }$$)+d\times f(\lambda_{1}$$,\;\lambda_{2}$$,\;\lambda_{3}$) Dermis: Hyperelastic Ogden model (α = 57.89) $\Phi =\left(2\mu /\alpha \right)(\lambda_{1}^{\alpha }$$+\lambda_{2}^{\alpha }$$+\lambda_{3}^{\alpha }$$)+d\;\times \;f(\lambda_{1}$$,\;\lambda_{2}$$,\;\lambda_{3}$) Hypodermis: Elastic	Stratum corneum: Elastic modulus (MPa): 0.752 Viable epidermis: Elastic modulus (MPa): 0.489 Dermis: Elastic modulus (MPa): 7.33 Hypodermis: Elastic modulus (Pa): 3.4 × 10^4^	A non-linear finite element model was established, the failure criterion was combined with the eroding surface-to-surface contact method to analyze the rupture of the skin.	[90]
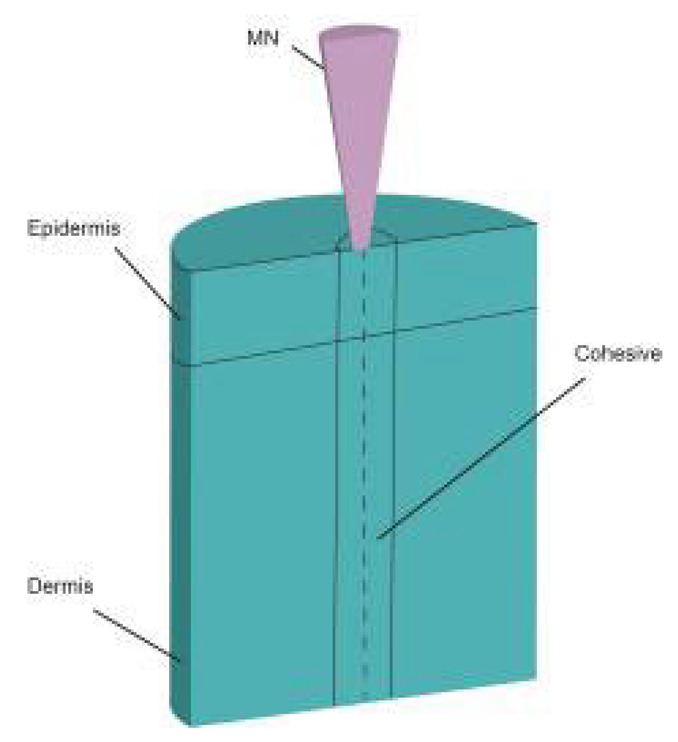	Epiderm: Ogden model Dermis: Ogden model Material failure criterion: Cohesive method G_1_ + (G_2_ − G_1_)(2G_2_/(G_1_ + G_2_))^ƞ^ = Gc	Epiderm: α: 2.9814; μ: 4.0991 Dermis: α: 3.2876; μ: 0.0226	Using the cohesive model and energy-based method to predict the path of skin injury and the contact between microneedles. It can be evaluated without defining life and death units.	[91]
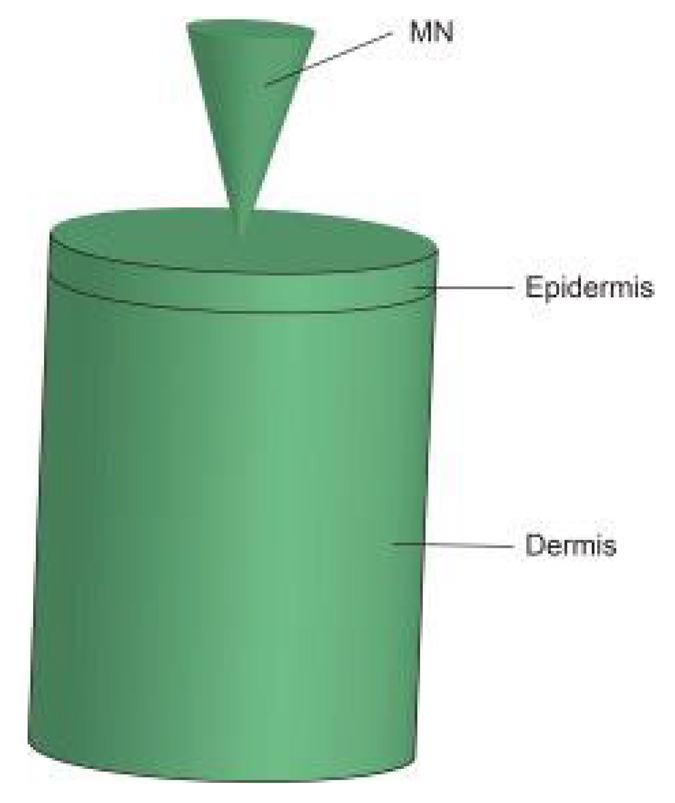	Epidermis: Elastic Dermis: Elastic	Epidermis: Elastic modulus(MPa): 1;υ: 0.495 Dermis: Elastic modulus(MPa): 0.066; υ: 0.495	The skin was defined as a linear elastic material, which can not predict skin damage and failure.	[44]
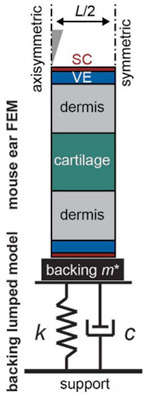	Stratum corneum: Hyperelastic Ogden model Viable epidermis: Hyperelastic Ogden model Dermis: Hyperelastic Ogden model	Stratum corneum: Elastic modulus (MPa): 3.35; Stretch exponent α: 5.77 Viable epidermis: Elastic modulus (MPa): 2.7 Stretch exponent α: 27.6 Dermis: Elastic modulus (MPa) = 27.4 Stretch exponent α: 15.5	It can simulate the damaged characteristics of the skin, describe the fractured image, and predict the fracture depth. However, this model does not appear to be appropriate due to different behavior of the skin and ductile materials.	[88]
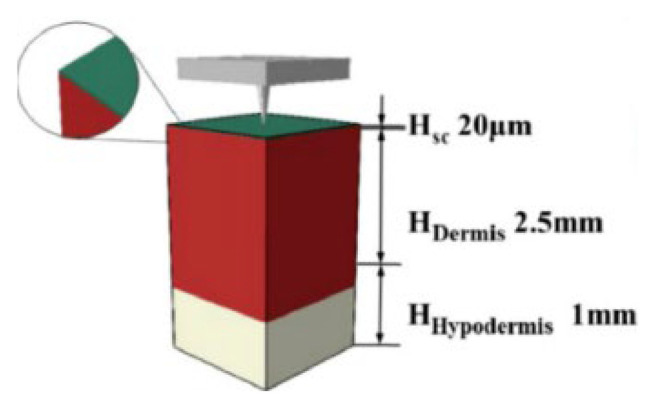	Stratum corneum: Neo-Hookean Dermis: Gasser-Ogden-Holzapfel Hypodermis: Linear elastic material	Stratum corneum: $\mathrm{Stiffness},\;C_{10}\left(\mathrm{MPa}\right)$:10; Compressibility value, D_1_: 1.03 × 10^−7^ Dermis: µ (MPa)$:\;0.1007;\;k_{1}\left(\mathrm{MPa}\right)$: 24.53; k_2_: 0.1327; Compressibility value, D_1_: 1.03 × 10^−7^ Hypodermis: Young’s modulus(KPa): 34 Poisson’s ratio: 0.48	A two-analysis step (Skin Stretching-Microneedle Penetration) was employed,, which can provide a quantitative and detailed analysis of the microneedle-skin interaction. However, mesh dependency is a major challenge.	[92]

## Data Availability

Not applicable.

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
