# Peer review of "The Finite Element Analysis Research on Microneedle Design Strategy and Transdermal Drug Delivery System"

_pharmaceutics, 2022, doi:10.3390/pharmaceutics14081625_

Round 1

Reviewer 1 Report

My comments on this article are quite brief. There are two main points: as a descriptive article of this field it is very good and provides a solid basis to learning about this field. As such, it is a good piece of work that will gain significant readership.

However, the work is almost completely devoid of any critical element and reads more like a book chapter than a review. To be a useful article for a broad market would require the authors to be more critical of their field. For example, they should discuss more explicitly and critically the failure of these products to be clinically relevant, and then to understand how that failure has shaped the field itself. Further, the impact of design and computer-based or rational design methods is not systematically reported or discussed, it is simply presented as chronologically linear prose. 

So the article should present itself with greater critical context and then place their successes and failures of this field in the correct frame for the reader to understand the main issues in it

Author Response

Response to Reviewer 1 Comments

Comment 1: My comments on this article are quite brief. There are two main points: as a descriptive article of this field it is very good and provides a solid basis to learning about this field. As such, it is a good piece of work that will gain significant readership.

However, the work is almost completely devoid of any critical element and reads more like a book chapter than a review. To be a useful article for a broad market would require the authors to be more critical of their field. For example, they should discuss more explicitly and critically the failure of these products to be clinically relevant, and then to understand how that failure has shaped the field itself. Further, the impact of design and computer-based or rational design methods is not systematically reported or discussed, it is simply presented as chronologically linear prose.

So the article should present itself with greater critical context and then place their successes and failures of this field in the correct frame for the reader to understand the main issues in it.

Response: Thank you for pointing this out. We have already added some content in the review and discussed the clinically related challenges of the microneedle technology to help the readers better understand the field.

Reviewer 2 Report

Dear authors

The work you've presented is an interesting review about how to apply the finite element methodology to microneedles for skin. Except one misleading allocation of fig 8 that, surely Editors will repare, the rest of text, in my opinion is worthy to be published

Many thanks

Author Response

Response to Reviewer 2 Comments

Comment 2: The work you've presented is an interesting review about how to apply the finite element methodology to microneedles for skin. Except one misleading allocation of fig 8 that, surely Editors will repare, the rest of text, in my opinion is worthy to be published。

Response: We apologize for the confusion. The manuscript has been revised accordingly. The order of Figures 8 and 7 have been corrected.

Reviewer 3 Report

In this manuscript, Yan et al. introduces FEA research for microneedle transdermal drug delivery system, focusing on microneedles design strategy, skin mechanics models, skin permeability, and the FEA research on drug delivery by MNs. This is an interesting manuscript, but a few mechanisms are unclear in the current format, which I have listed below;

- line 18 - 'Many research work....' should be 'Many research works ...'.

- Line 40 - 'I.V. administration' I.V should be spelled out. All other abbreviations should be corrected as well, they must be spelled out when they have been used in the first place.

- Introduction: 'a very brief additions should be on transdermal biosensing as well, where microneedles have extensively been used. Authors must cite the paper;  https://doi.org/10.1002/wnan.1699

- Introduction, a little bit detail should be added on clinical challenges of the microneedle technology.

- Figure 5: the legend should be extended, it is not clear what is going on in the figure. Furthermore, the permission to reproduce the figures should be added too.

- Same applies to figure 6.

- Same applies for figure 8 and other figures. I cannot permission rights etc. please make sure this has been properly addressed.

- The limitations and promises should be extended in the conclusion section.

Author Response

Response to Reviewer 3 Comments

Comment 3: In this manuscript, Yan et al. introduces FEA research for microneedle transdermal drug delivery system, focusing on microneedles design strategy, skin mechanics models, skin permeability, and the FEA research on drug delivery by MNs. This is an interesting manuscript, but a few mechanisms are unclear in the current format, which I have listed below;

- line 18 - 'Many research work....' should be 'Many research works ...'.

- Line 40 - 'I.V. administration' I.V should be spelled out. All other abbreviations should be corrected as well, they must be spelled out when they have been used in the first place.

- Introduction: 'a very brief additions should be on transdermal biosensing as well, where microneedles have extensively been used. Authors must cite the paper;  https://doi.org/10.1002/wnan.1699

- Introduction, a little bit detail should be added on clinical challenges of the microneedle technology.

- Figure 5: the legend should be extended, it is not clear what is going on in the figure. Furthermore, the permission to reproduce the figures should be added too.

- Same applies to figure 6.

- Same applies for figure 8 and other figures. I cannot permission rights etc. please make sure this has been properly addressed.

- The limitations and promises should be extended in the conclusion section.

Response: For the problem of line 18 and Line 40, we apologize for the confusion. The manuscript has been revised accordingly. For the introduction, thank you for pointing this out, and we have enriched the introduction by adding the clinical challenges details of this technology. The paper “https://doi.org/10.1002/wnan.1699” was also appropriately cited and added to the references. For the copyright issue of the pictures, we have been applying for the permission. The legend were also extended. The limitations and promises were extended in the conclusion section as well.

Round 2

Reviewer 3 Report

I am pleased to recommend the revised manuscript for publication in Pharmaceutics. 

Author Response

Thanks very much for taking your time to review this manuscript. I really appreciate all your comments and suggestions!